# Feedback optimizes neural coding and perception of natural stimuli

**Chengjie G Huang[†], Michael G Metzen[†], Maurice J Chacron\***

Department of Physiology, McGill University, Montreal, Canada

**Abstract** Growing evidence suggests that sensory neurons achieve optimal encoding by matching their tuning properties to the natural stimulus statistics. However, the underlying mechanisms remain unclear. Here we demonstrate that feedback pathways from higher brain areas mediate optimized encoding of naturalistic stimuli via temporal whitening in the weakly electric fish *Apteronotus leptorhynchus*. While one source of direct feedback uniformly enhances neural responses, a separate source of indirect feedback selectively attenuates responses to low frequencies, thus creating a high-pass neural tuning curve that opposes the decaying spectral power of natural stimuli. Additionally, we recorded from two populations of higher brain neurons responsible for the direct and indirect descending inputs. While one population displayed broadband tuning, the other displayed high-pass tuning and thus performed temporal whitening. Hence, our results demonstrate a novel function for descending input in optimizing neural responses to sensory input through temporal whitening that is likely to be conserved across systems and species.

DOI: https://doi.org/10.7554/eLife.38935.001

**\*For correspondence:**
maurice.chacron@mcgill.ca

[†]These authors contributed equally to this work

**Competing interests:** The authors declare that no competing interests exist.

## Introduction

How sensory neurons process incoming sensory input thereby leading to perception and behavior remains a central question in systems neuroscience. There is growing evidence showing that neural systems can efficiently process natural sensory input by matching their tuning properties to natural stimulus statistics, thereby removing redundancy and thus maximizing information transmission (*Fairhall et al., 2001*; *Brenner et al., 2000*; *Maravall et al., 2007*). Theory posits that efficient coding is achieved by ensuring that the neural tuning function is inversely proportional to stimulus intensity as a function of frequency, thereby achieving a neural response whose amplitude is independent of frequency (*Rieke et al., 1996*). While such 'whitening' has been observed across species and systems (*Dan et al., 1996*; *Wang et al., 2003*; *Huang et al., 2016*; *Pozzorini et al., 2013*; *Pitkow and Meister, 2012*), the nature of the underlying mechanisms remains unclear.

Weakly electric fish offer an attractive model system for studying the mechanisms underlying optimized coding of natural stimuli because of well-characterized anatomy and physiology (*Clarke et al., 2015*; *Berman and Maler, 1999*; *Chacron et al., 2011*). These fish generate a quasi-sinusoidal signal called the electric organ discharge (EOD) around their body, thereby allowing them to explore the environment and communicate with conspecifics. When two conspecifics are located close (<1 m) to one another, each fish experiences an amplitude modulation of its own EOD (i.e., a beat or first-order) whose amplitude (i.e., envelope or second-order) is a function of the distance and relative orientation between two conspecifics (*Yu et al., 2012*; *Fotowat et al., 2013*) (see (*Stamper et al., 2013*) for review). Natural electrosensory envelopes due to movement measured in freely moving animals display scale invariance in that their spectral power decays as a power law as a function of increasing temporal frequency (*Fotowat et al., 2013*; *Metzen and Chacron, 2014*). Weakly electric fish furthermore give robust behavioral responses to movement related envelopes in which the EOD frequency tracks the stimulus' detailed (*Metzen and Chacron, 2014*).

Envelopes and other electrosensory stimuli are sensed by peripheral receptors scattered over the animal's skin. These project to pyramidal cells within the electrosensory lateral line lobe (ELL) which in turn project to higher brain structures, thereby giving rise to behavior (*Figure 1B*). ELL pyramidal cells also receive large amounts of descending inputs (i.e., feedback) from higher brain centers such as the nucleus praeeminentialis (nP) and contact ELL pyramidal cells either directly (i.e., the direct feedback pathway) or indirectly (i.e., the indirect feedback pathway; *Figure 1B*) (*Berman and Maler, 1999*; *Sas and Maler, 1983*; *Sas and Maler, 1987*). Pyramidal cell responses to beats and other first-order electrosensory stimuli are well characterized in general (see (*Clarke et al., 2015*; *Chacron et al., 2011*; *Marsat et al., 2012*; *Krahe and Maler, 2014*; *Huang and Chacron, 2017*) for review) and several studies have shown that both direct and indirect feedback inputs play important roles in shaping (*Bastian, 1986a*; *Bastian et al., 2004*; *Bastian, 1999*; *Bol et al., 2011*; *Chacron, 2006*; *Chacron et al., 2005*) as well as synthesizing (*Clarke and Maler, 2017*) these. Recent studies have focused on characterizing ELL pyramidal cell responses to envelopes (*Huang et al., 2016*; *Stamper et al., 2013*; *Huang and Chacron, 2017*; *Middleton et al., 2006*; *Zhang and Chacron, 2016*; *Huang and Chacron, 2016*; *Martinez et al., 2016*). Specifically, it was shown that ELL pyramidal cells can optimally encode natural electrosensory envelopes because their high-pass tuning properties effectively oppose the decaying stimulus power, such that the response power is whitened (*Huang et al., 2016*; *Zhang and Chacron, 2016*; *Huang and Chacron, 2016*; *Martinez et al., 2016*). However, whether and, if so, how direct and indirect feedback inputs mediate optimized coding of envelopes by ELL pyramidal cells has not been investigated to date.

Here we used a systems level approach to investigate how feedback input mediates optimized processing of natural envelope stimuli by ELL pyramidal cells. Pharmacological inactivation of all sources of descending input strongly reduced pyramidal cell and behavioral responses to envelopes. However, pyramidal cell responses to high envelope frequencies were relatively more attenuated than those to low envelope frequencies, such that the resulting tuning was no longer high-pass but independent of envelope frequency. As a consequence, optimized coding of natural stimuli by temporal whitening was compromised. In contrast, selective inactivation of indirect feedback input strongly increased both pyramidal cell and behavioral sensitivity to envelopes. However, enhancement was primarily seen for low envelope frequencies, such that the resulting pyramidal cell tuning curve was broadband, which also compromised optimized coding of natural stimuli by temporal whitening. Finally, we recorded from two different groups of nP neurons that project either directly or indirectly back to ELL pyramidal cells. nP neurons projecting directly to ELL displayed broadband envelope frequency tuning curves and thus did not perform temporal whitening. In contrast, nP neurons projecting indirectly to ELL displayed high-pass envelope frequency tuning curves that effectively opposed the decaying envelope stimulus spectral power content, thereby enabling temporal whitening. Our results demonstrate clear but distinct functional roles for both direct and indirect feedback pathways in determining how ELL pyramidal cells respond to envelopes. While the direct pathway enhances responses to envelopes independently of frequency, the indirect pathway instead selectively attenuates responses to low frequencies, thereby giving rise to the high-pass tuning that is necessary to optimize coding of natural envelopes via temporal whitening. Interestingly, our results also show that indirect feedback input to ELL is temporally whitened. Our results thus demonstrate an important new function for descending inputs onto sensory neurons in optimizing their responses to natural stimuli and their perception at the organismal level.

## Results

We investigated the mechanisms that enable ELL pyramidal cells to optimally encode envelope stimuli. To do so, we performed recordings from these in awake behaving animals (*Figure 1A*). Data obtained from ON- and OFF-type ELL pyramidal neurons were pooled because, consistent with previous studies (*Huang and Chacron, 2016*), we found no overall difference between their responses to envelopes. Our stimuli consisted of sinusoidal AMs with constant amplitude as well as noisy EOD AMs whose envelope varied sinusoidally at different frequencies. The left panel of *Figure 1A* shows example traces of the AM (blue, first-order), envelope (red, second-order), and the full signal (cyan) received by the animal with their respective temporal frequency contents. It is further important to realize that the animal's EOD is a carrier and that the meaningful stimulus here is the EOD AM. Thus, both first- and second-order features of the stimulus correspond to the second- and third-

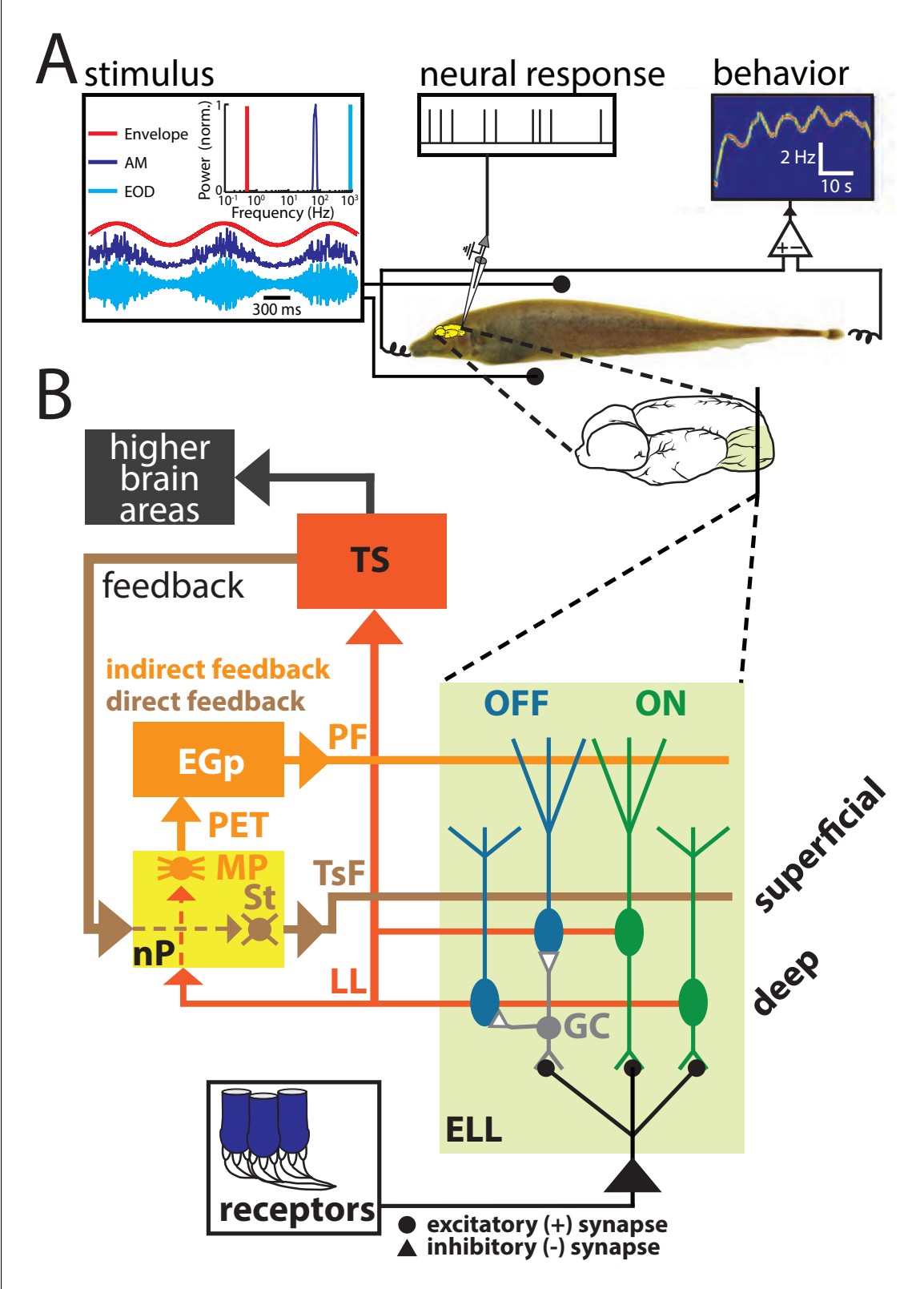

**Figure 1.** (A) Schematic of the experimental setup showing the awake-behaving preparation where a stimulus (left) is presented to the animal while neural (upper middle) and behavioral (upper right) responses are recorded simultaneously. The stimuli consisted of amplitude modulations of the animal's own EOD: shown are an example AM waveform (blue), its envelope (red), and the full signal received by the animal (cyan) with their respective frequency contents. (B) Simplified schematic showing the relevant anatomy and circuitry of the electrosensory system. Peripheral receptors make

*Figure 1 continued*

excitatory synaptic contact with ON-type pyramidal cells (green) within the electrosensory lateral line lobe (ELL) and local interneurons (granule cells: GC) that inhibit OFF-type pyramidal cells (blue). All ELL pyramidal cells project to the midbrain torus semicircularis (TS) via the lateral lemniscus (LL), which in turn projects to higher brain areas (black) but also projects back to the nucleus praeeminentialis (nP). Stellate cells (St) within nP receive input from TS and project back to ELL pyramidal cells via the tractus stratum fibrosum (TsF). This feedback loop is known as the direct feedback pathway. A subclass of ELL pyramidal cells (deep cells) projects to nP multipolar cells (MP) that, together with several other cell types within nP (not shown) (*Sas and Maler, 1983*; *Sas and Maler, 1987*), in turn project to the Eminentia Granularis posterior (EGp) via the praeeminentialis electrosensory tract (PET). EGp granule cells (not shown) project back to ELL via parallel fibers (PF). This feedback loop is known at the indirect feedback pathway.

DOI: https://doi.org/10.7554/eLife.38935.002

order features of the full signal received by the animal, respectively. We recorded both the neural activity as well as the animal's behavioral response that consists of changes in the EOD frequency (*Figure 1A*). *Figure 1B* shows a simplified diagram of the relevant anatomy and circuitry of the electrosensory system. Peripheral receptors make synaptic contact onto ELL pyramidal cells either directly with excitation for ON-type cells or indirectly via local inhibitory interneurons for OFF-type cells. Pyramidal cells are the sole output neurons of the ELL and project to the midbrain torus semicircularis (TS) which in turn projects to higher brain areas. As mentioned above, ELL pyramidal cells receive large amounts of feedback input both directly from nP (direct feedback; *Figure 1B*, brown) and indirectly via the Eminentia Granularis posterior (EGp) (indirect feedback; *Figure 1B*, orange). As explained above, previous studies have shown that ELL pyramidal cells can optimally encode natural envelope stimuli because their high-pass tuning curves (*Figure 2*, middle panel) are set to counter the decaying stimulus spectral power (*Figure 2*, left panel). The resulting response spectrum is thus independent of frequency (*Figure 2*, right panel) which optimizes information transmission (*Huang et al., 2016*; *Huang and Chacron, 2016*) (see (*Huang and Chacron, 2017*) for review).

## Descending input shapes neural responses and perception of envelopes

We performed several manipulations that either completely or selectively inactivated feedback input onto ELL pyramidal cells that are schematized in *Figure 3*. The first consists of completely inactivating feedback by injecting the sodium channel antagonist lidocaine bilaterally into nP (*Figure 3A*). The second consists of selectively inactivating the indirect feedback by injecting lidocaine bilaterally (for behavior) or ipsilaterally (for neurons) into PET (*Figure 3B*). The third consists of selectively inactivating the indirect feedback by injecting the non-competitive glutamate receptor antagonist 6-cyano-7-nitroquinoxaline-2,3-dione (CNQX) into the ELL molecular layer (*Figure 3C*). We note that injecting lidocaine into PET will block indirect feedback input to most if not all pyramidal cells within the ipsilateral ELL. In contrast, injecting CNQX into the ELL molecular layer will only block indirect feedback input in the vicinity of the ELL being recorded from (*Chacron et al., 2005*; *Bastian, 1993*).

We initially investigated the effects of complete feedback inactivation on neural and behavioral responses to envelopes (*Figure 4*). We found that complete feedback inactivation strongly attenuated ELL pyramidal neural responses to envelopes (*Figure 4A*, compare black and purple). Under control conditions, cells typically responded vigorously to the sinusoidal envelope (*Figure 4A*, top

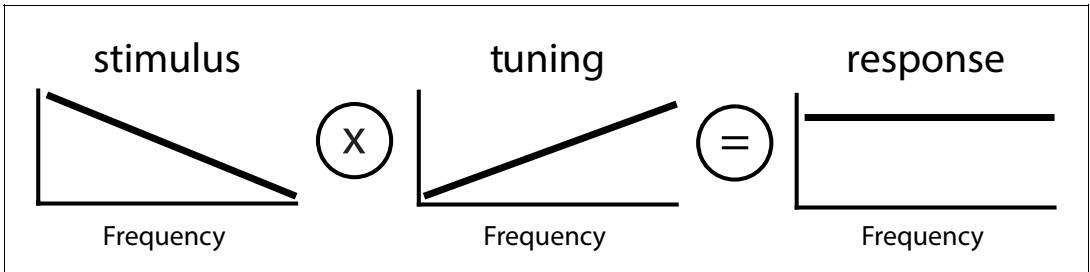

**Figure 2.** Principle of whitening by which the neural tuning curve (center) increases in order to effectively compensate for the decaying power spectrum of natural envelope stimuli (left), such that the resulting response power is independent of frequency (i.e., 'whitened', right).

DOI: https://doi.org/10.7554/eLife.38935.003

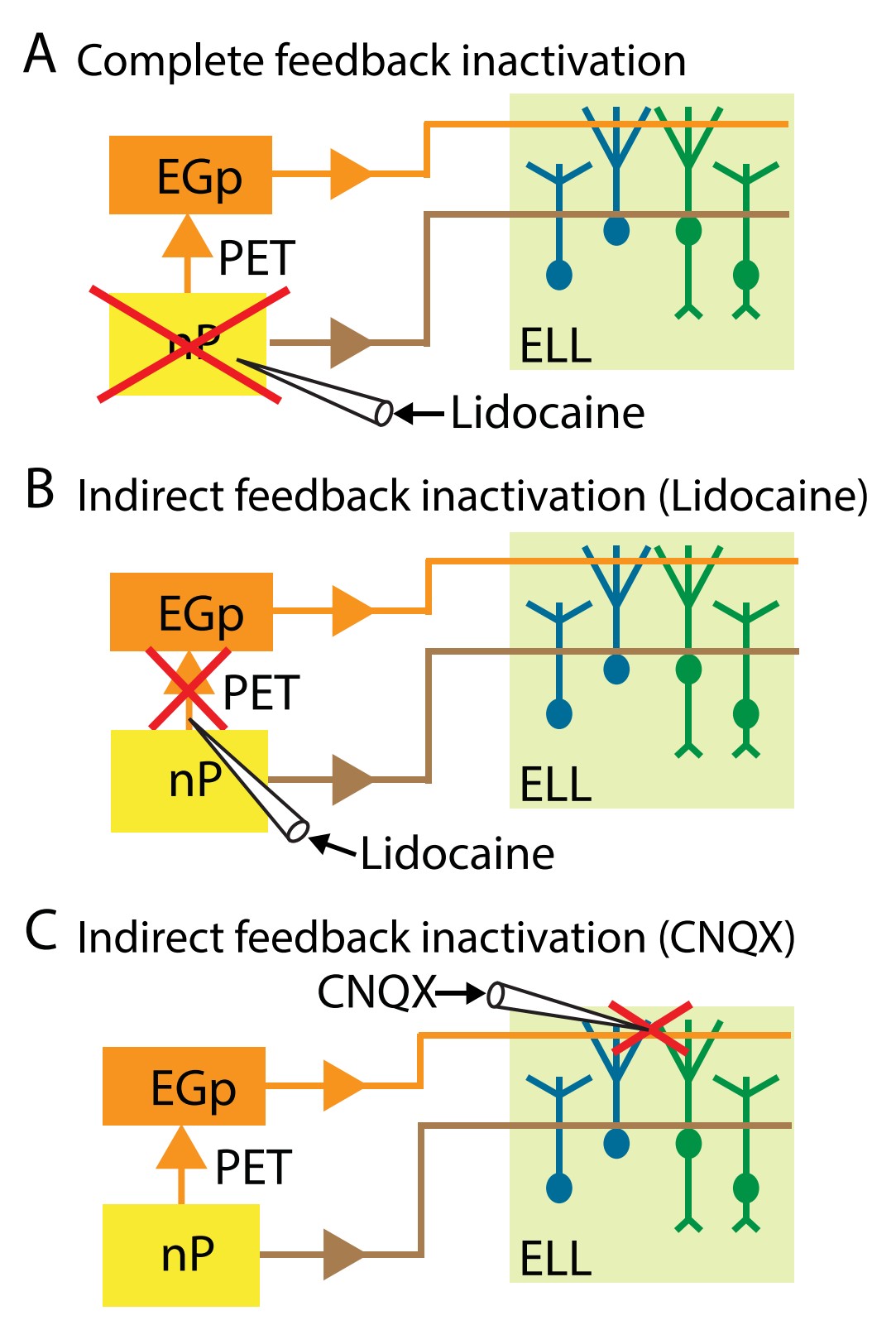

**Figure 3.** Summary of the different manipulations that were performed to either completely or selectively inactivate feedback input onto ELL pyramidal cells. In all cases, neural recordings were obtained from pyramidal cells within the ipsilateral ELL. (**A**) Schematic showing a method to inactivate both the direct and the indirect feedback pathways (i.e., complete feedback inactivation) that involves injecting lidocaine bilaterally into nP. (**B**) Schematic

*Figure 3 continued on next page*

*Figure 3 continued*

showing a method to inactivate the indirect feedback that involves injecting lidocaine bilaterally (for behavior) or ipsilaterally (for neurons) into PET. (**C**) Schematic showing an alternative method to inactivate the indirect feedback that involves injecting CNQX into the ipsilateral ELL molecular layer.

DOI: https://doi.org/10.7554/eLife.38935.004

panel, red) through changes in firing rate (*Figure 4A*, bottom panel, black). However, after complete feedback inactivation, responses to the envelope were strongly reduced (*Figure 4A*, bottom panel, purple). We next varied the envelope frequency and investigated the effects of complete feedback inactivation on envelope tuning. Under control conditions, the tuning curve of ELL pyramidal cells is high-pass, such that neural gain increases as a power law when envelope frequency is increased (*Figure 4B*, black). Further, the power law exponent is set such as to oppose the decay of the spectrum of natural envelopes, thereby causing the response power to be independent of frequency (i. e., is 'whitened', *Figure 4C*, black) as quantified by a white index (see methods) value near unity (*Figure 4F*, left panel, black), which is required for optimal coding. Complete feedback inactivation strongly affected tuning curves as well as temporal whitening. Indeed, we observed a strong attenuation in neural gain for all envelope frequencies tested (*Figure 4B*, compare black and purple). However, the attenuation was more pronounced for higher envelope frequencies, such as the resulting tuning curve was flat as characterized by a power law exponent near zero (*Figure 4B*, purple, see inset). Such a change in tuning strongly affected whitening as response power was no longer independent of frequency (*Figure 4C*, purple), as quantified by a lower white index value (*Figure 4F*, left panel, purple) together with a decrease in neural sensitivity (*Figure 4F*, right panel) indicating suboptimal coding. These results imply that feedback inputs optimize neural coding of envelopes by enhancing neural responses in a frequency dependent manner. Indeed, higher envelope frequencies are more amplified relative to lower envelope frequencies, thereby whitening neural responses to natural envelopes.

Changes in neural responses are only behaviorally relevant if they are actually decoded by downstream areas. Thus, we next investigated the effects of pharmacological inactivation of feedback pathways on behavioral responses. Under control conditions, the animal's EOD frequency tracks the envelope (*Figure 4D*, black). However, changes in EOD frequency are greater for low envelope frequencies, resulting in a behavioral gain that decreases as a power law for increasing envelope frequency, in a power-law manner (*Figure 4E*, black), consistent with previous results (*Metzen and Chacron, 2014*). Complete feedback inactivation strongly attenuated the animal's behavioral responses to envelopes (*Figure 4D*, purple) in a frequency dependent manner, such that attenuation was strongest for low envelope frequencies (*Figure 4E*, purple). Decreases in neural sensitivity were accompanied by decreases in behavioral sensitivity (*Figure 4F*, right panel).

Interestingly, complete feedback inactivation did not affect ELL pyramidal cell responses to AMs (*Figure 4—figure supplement 1*). We note that performing a 'Sham complete feedback inactivation' by injecting saline instead of lidocaine bilaterally into nP did not alter neural or behavioral responses to envelopes (*Figure 4—figure supplement 2*). Finally, our results showing that complete feedback inactivation strongly affects ELL neural responses to envelopes were robust. This is because injecting lidocaine unilaterally within the contralateral nP while recording from pyramidal cells within the ipsilateral ELL gave rise to similar changes in neural responses to envelopes (*Figure 4—figure supplement 3*).

## Direct feedback enhances while indirect feedback optimizes neural responses to natural envelopes

We next investigated how direct and indirect sources of descending input contribute to determining ELL pyramidal cell and behavioral responses to envelopes (*Figure 5*). To do so, we inactivated the indirect feedback pathway by injecting lidocaine into PET bilaterally (*Figure 3B*). Consistent with previous results, lidocaine injection increased ELL pyramidal neuron responses to AMs (*Figure 5—figure supplement 1*). We found that indirect feedback inactivation enhanced neural responses to envelopes (*Figure 5A*, compare black and orange). However, the envelope tuning quantified by neural gain was increased for low but not for high envelope frequencies, such that the tuning curve, which was initially high-pass (*Figure 5B*, black), became independent of frequency after lidocaine

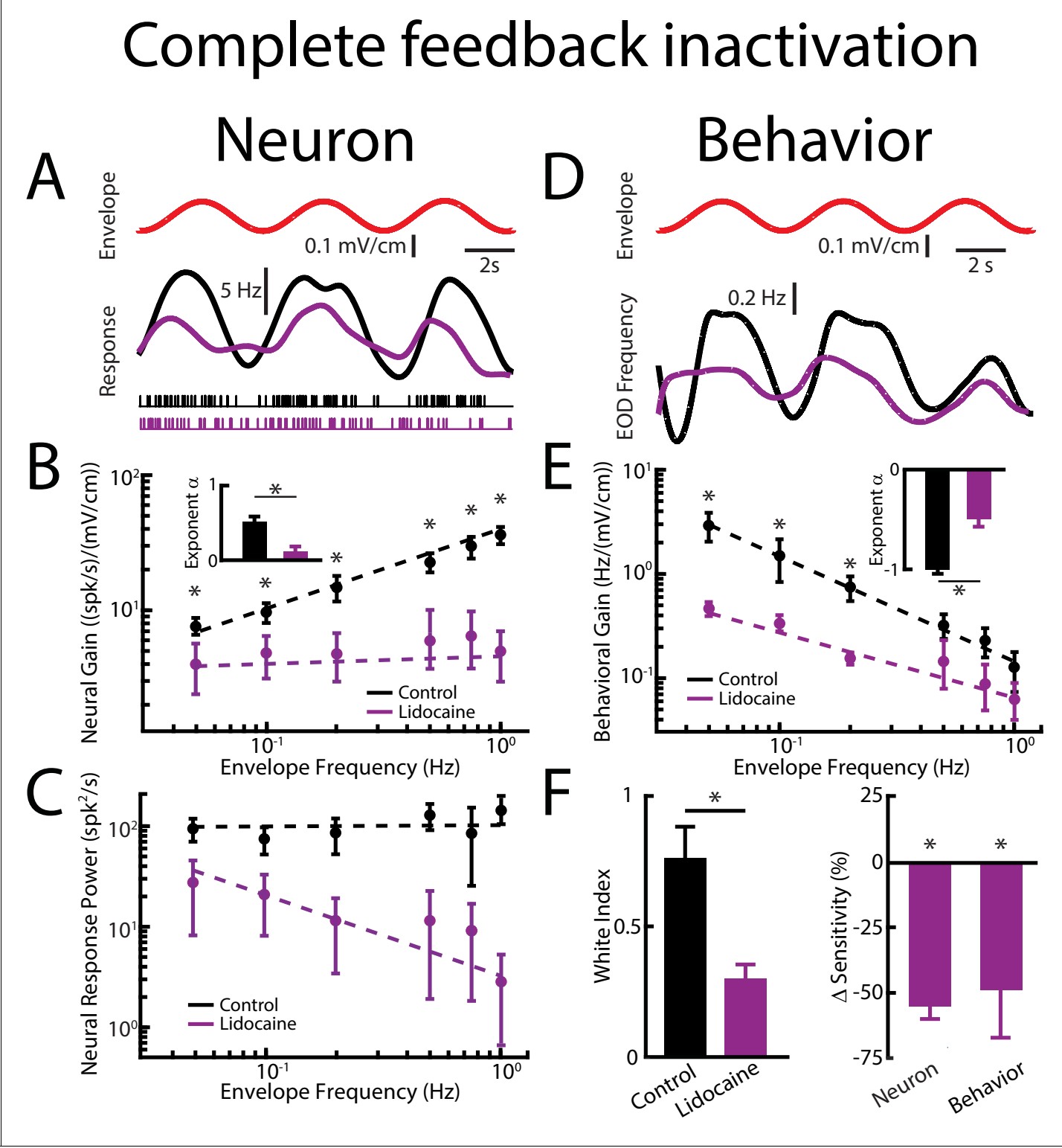

**Figure 4.** Feedback input enhances and optimizes information transmission via whitening. Results are shown before and after complete feedback inactivation was achieved via injection of lidocaine into nP. (**A**) *Top*: sinusoidal envelope waveform (red). *Middle*: time dependent firing rate from a typical ELL pyramidal cell before (black) and after (purple) lidocaine application. *Bottom*: spiking activity from this same neuron in response to stimulation before (black) and after (purple) lidocaine application. (**B**) Population-averaged tuning curve quantified by the neural gain as a function of envelope frequency before (black) and after (purple) lidocaine application. The dashed lines show the best power law fits to the data. Inset: Population-averaged best-fit power law exponents before (black) and after (purple) lidocaine injection were significantly different from one another (p=0.0156,

*Figure 4 continued on next page*

*Figure 4 continued*

Wilcoxon Signed-Rank Test). (C) Population-averaged neural response power as a function of envelope frequency before (black) and after (purple) lidocaine application. The dashed lines show the best power law fits to the data. (D) *Top*: sinusoidal envelope waveform (red). *Bottom*: time dependent EOD frequency from a typical fish before (black) and after (purple) lidocaine application. (E) Population-averaged behavioral gain as a function of envelope frequency before (black) and after (purple) lidocaine application. The dashed lines show the best power law fits to the data. Inset: Population-averaged best-fit power law exponents before (black) and after (purple) lidocaine injection were significantly different from one another (p=0.0313, Wilcoxon Signed-Rank Test). (F) *Left*: population-averaged white index values before (black) and after (purple) lidocaine application were significantly different from one another (p=0.0234, Wilcoxon Signed-Rank Test). *Right*: population-averaged relative changes in neural (left) and behavioral (right) gain following lidocaine application were both significantly different from zero (neuron: p=$4.77 \times 10^{-4}$, Wilcoxon Signed-Rank Test, behavior: p=0.002, Wilcoxon Signed-Rank Test). '*' indicates statistical significance at the p=0.05 level.

DOI: https://doi.org/10.7554/eLife.38935.005

The following figure supplements are available for figure 4:

**Figure supplement 1.** Complete feedback inactivation does not affect ELL pyramidal cell responses to AMs.
DOI: https://doi.org/10.7554/eLife.38935.006

**Figure supplement 2.** Sham complete feedback inactivation achieved by injecting saline bilaterally into nP has no effect on behavior and ELL pyramidal cell tuning properties, as well as optimized coding of natural stimuli.
DOI: https://doi.org/10.7554/eLife.38935.007

**Figure supplement 3.** Contralateral feedback inactivation achieved by injecting lidocaine into the contralateral nP gives rise to effects qualitatively similar to those observed when injecting lidocaine bilaterally when recording from pyramidal cells within the ipsilateral ELL.
DOI: https://doi.org/10.7554/eLife.38935.008

injection (*Figure 5B*, orange). This change in tuning antagonized optimal coding of envelopes, in that the response power spectrum was no longer independent of envelope frequency (*Figure 5C*, compare black and orange), as quantified by a decrease in the white index (*Figure 5F*, left, orange). Neural sensitivity was significantly increased (*Figure 5F*, right, orange). Thus, our results show that indirect sources of descending input actively shape ELL pyramidal cell tuning to envelopes by attenuating responses to low but not high envelope frequencies, thereby optimizing coding through whitening.

Indirect feedback inactivation significantly enhanced the animal's behavioral responses to envelopes (*Figure 5D*, compare black and orange) only for low frequencies (*Figure 5E*, compare black and orange). The behavioral response curve thus decreased more steeply with increasing envelope frequency as reflected by a significant decrease in the behavioral exponent (*Figure 5E*, inset), leading to an increased behavioral sensitivity to envelopes only at low frequencies (*Figure 5F*, right, orange). Taken together, our results show that both direct and indirect feedback inputs onto ELL pyramidal cells have differential effects on envelope tuning and optimized coding. Specifically, they suggest that the function of the direct input is to enhance envelope responses independently of frequency while the indirect input selectively attenuates low frequencies, thereby optimizing coding.

We note that inactivating indirect feedback by injecting CNQX in the ELL molecular layer (*Figure 3C*) gave rise to similar effects on ELL pyramidal cell responses to envelopes (*Figure 5—figure supplement 2*). Also, consistent with previous results (*Bastian et al., 2004*), this manipulation significantly increased ELL pyramidal cell responses to AMs (*Figure 5—figure supplement 3*).

## Responses of nP neurons that give rise to descending input onto ELL pyramidal cells

Finally, we investigated the nature of the descending signals that are received by ELL pyramidal cells. To do so, we recorded from nP neurons that project both directly and indirectly to ELL (*Figure 6A*). Specifically, nP stellate cells project directly to ELL pyramidal cells while nP multipolar cells instead project indirectly through the EGp (*Sas and Maler, 1983*; *Sas and Maler, 1987*). Both cell types can easily be distinguished from one another based on their electrophysiological properties (*Bastian and Bratton, 1990*; *Bratton and Bastian, 1990*) (*Figure 6—figure supplement 1*). Overall, our results show that both stellate and multipolar cells responded strongly to envelopes (*Figure 6B*) but showed differential envelope frequency tuning (*Figure 6C*). Specifically, stellate cell sensitivity was largely independent of envelope frequency (*Figure 6C*, brown) as quantified by a power law exponent near zero (*Figure 6E*, left panel, brown). As such, these cells did not perform temporal whitening of natural envelopes as their response power spectra decayed with increasing

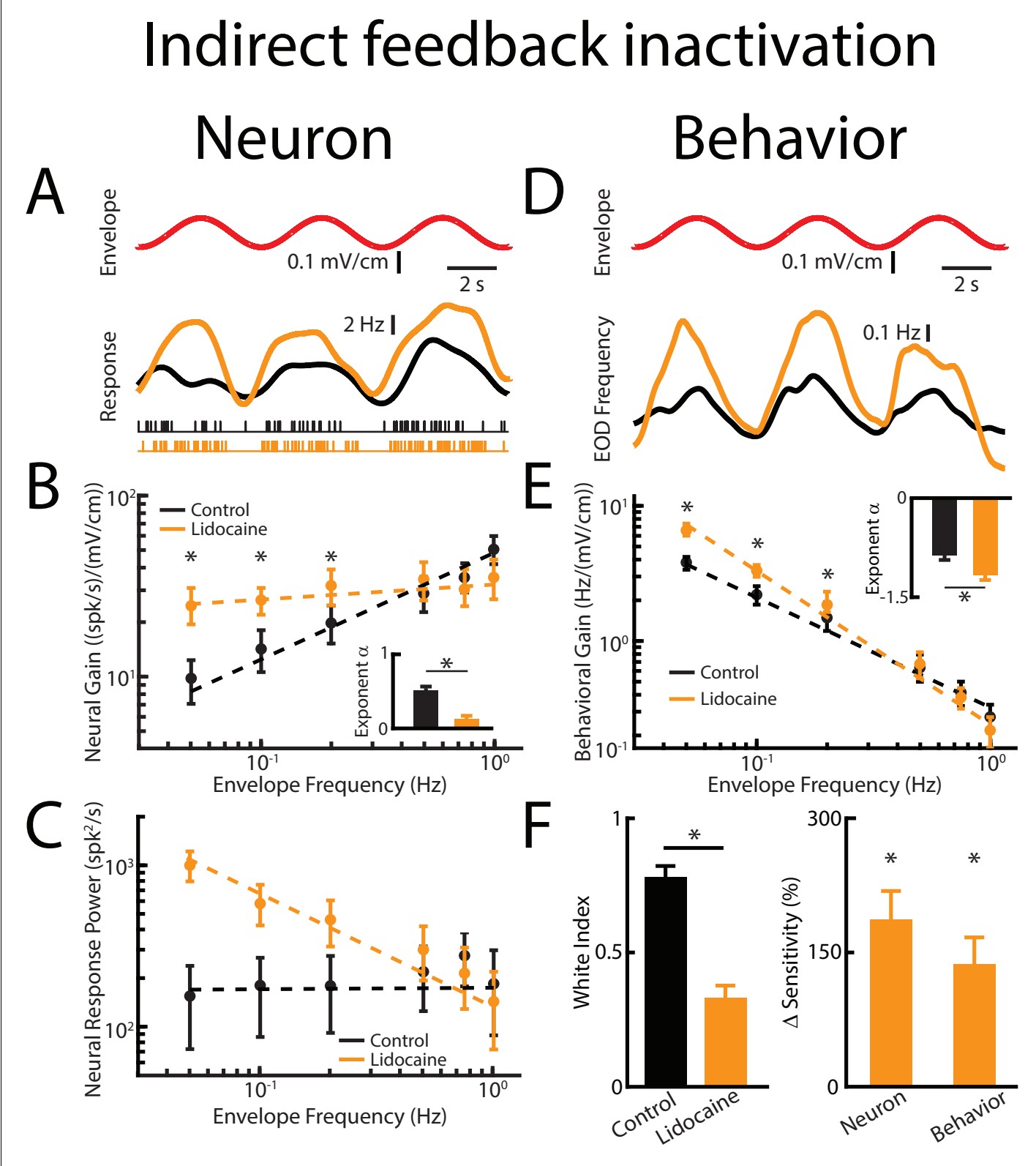

**Figure 5.** Direct feedback input enhances while indirect input optimizes neural responses. Results are shown before and after indirect feedback inactivation was achieved via bilateral injection of lidocaine into the PET. Data obtained from ELL pyramidal neurons were pooled as there are no significant differences between the envelope response of ON- and OFF-type pyramidal cells (*Huang and Chacron, 2016*). (**A**) *Top*: Sinusoidal envelope waveform (red). *Middle*: Time dependent firing rate from a typical ELL pyramidal cell before (black) and after (orange) lidocaine application. *Bottom*:
*Figure 5 continued on next page*

*Figure 5 continued*

spiking activity from this same neuron in response to stimulation before (black) and after (orange) lidocaine application. (B) Population-averaged tuning curve quantified by neural gain to sinusoidal envelopes as a function of envelope frequency before (black) and after (green) lidocaine application. The dashed lines show the best power law fits to the data. Inset: population-averaged best-fit power law exponent before (black) and after (orange) lidocaine injection (p=0.0039, Wilcoxon Signed-Rank Test). (C) Population-averaged neural response power as a function of envelope frequency before (black) and after (orange) lidocaine application. The dashed lines show the best power law fits to the data. (D) *Top*: sinusoidal envelope waveform (red). *Bottom*: Time dependent EOD frequency from a typical fish before (black) and after bilateral (orange) lidocaine injection. (E) Population-averaged behavioral gain as a function of envelope frequency before (black) and after (orange) lidocaine injection. Inset: population-averaged best-fit power law exponent before (black) and after (orange) lidocaine injection (p=0.0234, Wilcoxon Signed-Rank Test). The dashed lines show the best power law fits to the data. (F) *Left*: population-averaged white index before (black) and after (orange) lidocaine application (p=0.0273, Wilcoxon Signed-Rank Test). *Right*: population-averaged relative changes in neural and behavioral sensitivity following lidocaine application (neuron: $p=7.03*10^{-6}$, Wilcoxon Signed-Rank Test, behavior: $p=6.74*10^{-4}$, Wilcoxon Signed-Rank Test). '*' indicates statistical significance at the p=0.05 level.

DOI: https://doi.org/10.7554/eLife.38935.009

The following figure supplements are available for figure 5:

**Figure supplement 1.** Indirect feedback inactivation achieved by injecting lidocaine bilaterally into PET increases ELL pyramidal cell responses to AMs, consistent with previous results (*Bastian, 1986b*).

DOI: https://doi.org/10.7554/eLife.38935.010

**Figure supplement 2.** Indirect feedback inactivation achieved by injecting CNQX within the ELL molecular layer gives rise to effects on ELL pyramidal cell responses to envelopes that are qualitatively similar to those observed when injecting lidocaine into PET.

DOI: https://doi.org/10.7554/eLife.38935.011

**Figure supplement 3.** Indirect feedback inactivation achieved by injecting CNQX within the ELL molecular layer increases ELL pyramidal cell responses to AMs, consistent with previous results (*Bastian et al., 2004*; *Clarke and Maler, 2017*).

DOI: https://doi.org/10.7554/eLife.38935.012

frequency (*Figure 6D and E* right, brown). These results confirm our hypothesis that the function of the direct feedback input is to enhance ELL pyramidal cell responses to envelopes independently of temporal frequency.

In contrast, multipolar cells instead displayed high-pass tuning to envelopes (*Figure 6C*, orange) as quantified by a power law exponent near 0.4 (*Figure 6E* left, orange) that is similar to that observed for ELL pyramidal cells (compare with *Figure 4B*). As a result, we found that multipolar cells perform temporal whitening of envelopes as their response spectra was independent of frequency (*Figure 6D*, orange) as quantified by a white index near unity (*Figure 6E* right, orange). Thus, our results reveal that the feedback input that is sent indirectly to ELL pyramidal cells via the EGp is already temporally whitened. This result has important implications for understanding how temporal whitening of ELL pyramidal cell responses is achieved as discussed below.

## Discussion

We investigated the roles of both direct and indirect sources of descending input onto ELL pyramidal cells in determining their responses to envelopes. Pharmacological inactivation of both direct and indirect sources strongly attenuated pyramidal cell and behavioral responses to envelopes. Because responses to higher envelope frequencies were more attenuated, the resulting tuning curve became independent of frequency, thereby compromising optimized coding through temporal whitening. Pharmacological inactivation of indirect input instead increased pyramidal cell and behavioral responses to envelopes. However, enhancement was observed primarily for low envelope frequencies, such that the resulting tuning curve was independent of frequency, which also compromised optimized coding through temporal whitening. Finally, we investigated the nature of the feedback signals being received both directly and indirectly by ELL pyramidal cells. Specifically, nP stellate cells that project directly to ELL displayed tuning curves that were independent of envelope frequency and did not perform temporal whitening. In contrast, nP multipolar cells that project indirectly to ELL displayed high-pass tuning and optimally encoded envelopes through temporal whitening. Thus, our results provide the first experimental evidence showing how descending pathways mediate optimized coding of stimuli by sensory neurons. While direct feedback input enhances neural responses independently of frequency, our results show that indirect feedback input selectively attenuates responses to low envelope frequencies, thereby giving rise to a high-pass tuning that opposes natural envelope statistics and optimizes coding through temporal whitening.

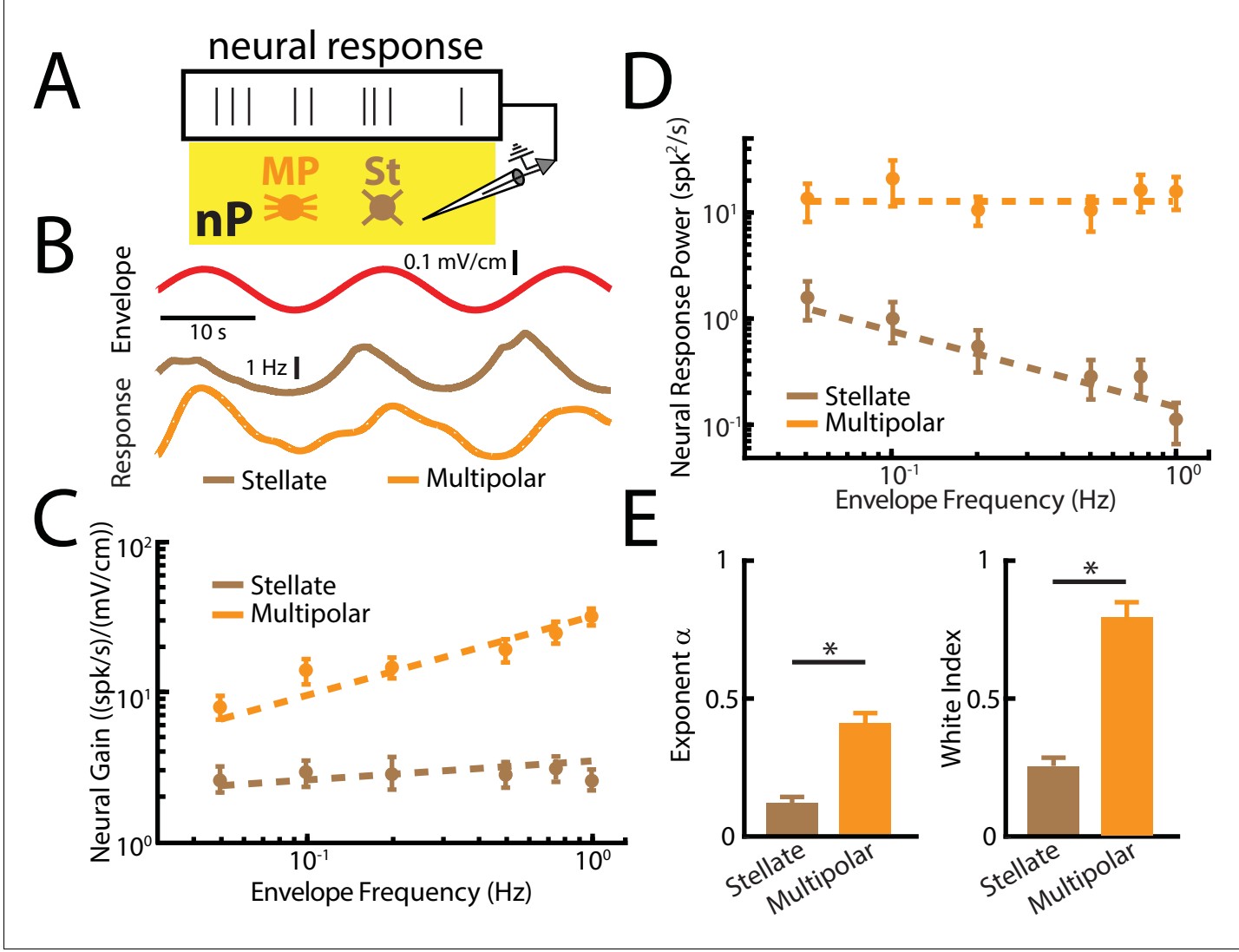

**Figure 6.** nP neurons projecting indirectly to ELL display tuning properties that are optimized to natural stimulus statistics. (A) Recordings were obtained from either Stellate (St) or multipolar (MP) cells within nP. (B) *Top*: sinusoidal envelope waveform (red). *Bottom*: Time dependent firing from typical nP stellate (brown) and multipolar (orange) cells. (C) Population-averaged tuning curve quantified by neural gain as a function of envelope frequency for nP stellate (brown) and multipolar (orange) cells. The dashed lines show the best power law fits to the data. (D) Population-averaged neural response power for stellate (brown) and multipolar (orange) cells. The dashed lines show the best power law fits to the data. (E) Population-averaged best-fit power law exponents (left) and white index (right) values for stellate (brown) and multipolar (orange) cells. In both cases, values obtained for stellate and multipolar cells were significantly different from one another (exponent: $\chi^2 = 12$, p=5.32*10$^{-4}$, Kruskal-Wallis ANOVA; white index: $\chi^2 = 10.7$, p=0.0011, Kruskal-Wallis ANOVA). '*' indicates statistical significance at the p=0.05 level.

DOI: https://doi.org/10.7554/eLife.38935.013

The following figure supplement is available for figure 6:

**Figure supplement 1.** Distinguishing between nP stellate and multipolar cells using previously characterized differences in their electrophysiological properties.

DOI: https://doi.org/10.7554/eLife.38935.014

Our results provide a new function for this feedback pathway by showing that nP stellate cells enhance the responses of ELL pyramidal cells to envelopes. Indeed, while previous studies have suggested that the function of this feedback pathway was to enhance responses to salient stimuli (*Berman and Maler, 1999*; *Bratton and Bastian, 1990*; *Berman and Maler, 1998*), experimental evidence supporting this hypothesis was lacking until recently when a clear role in synthesizing responses to motion stimuli consisting exclusively of first-order stimulus features was established

(*Clarke and Maler, 2017*). A recent study has furthermore shown that the direct feedback pathway enables neural responses to weak envelope stimuli (*Metzen et al., 2018*). Our results show an important novel functional role for the direct feedback pathway in enhancing both neural responses to and perception of behaviorally relevant second-order (i.e., envelope) stimuli that is independent of frequency. Interestingly, previous studies have suggested that the direct feedback pathway could function as a sensory searchlight (*Berman and Maler, 1999*), as originally proposed by Crick (*Crick, 1984*), thereby enhancing salient stimulus features and attenuating others. Our results are consistent with descending input onto ELL pyramidal cells acting in this manner but instead suggest that concerted action from both direct and indirect sources is necessary. Indeed, while the direct pathway enhances responses to envelopes independently of frequency, the indirect pathway instead selectively attenuates responses to low frequencies, thereby favoring responses to higher frequencies.

Overall, our results are consistent with previous ones showing that the tuning function of ELL pyramidal cells must be matched to natural statistics in order to optimize coding, which in turn ensures that behavioral sensitivity is greatest for frequencies at which natural stimuli contain the most power (*Huang et al., 2016*). However, we found that feedback inactivation altered ELL pyramidal cell tuning, thereby rendering coding sub-optimal, which caused a mismatch between behavioral sensitivity and natural statistics. However, our results suggest that the relationship between ELL pyramidal cell and behavioral responses to envelopes is more complicated than previously expected. Specifically, it was assumed that changes in the neural exponent of ELL pyramidal cells alone would determine changes in the behavioral exponent. Our current results show that this is not the case as pharmacological inactivation of both direct and indirect feedback input led to similar changes in the neural tuning exponent but led to opposite changes in the behavioral tuning exponent. Thus, it is not only the tuning exponent of ELL pyramidal cells that determines the behavioral exponent, but also the overall sensitivity. Further studies are needed to better understand how information transmitted by ELL pyramidal cells is decoded by higher brain areas in order to lead to behavioral responses.

Previous results have demonstrated important functions for the indirect feedback pathway such as gain control (*Bastian, 1986a*; *Bastian, 1986b*) as well as cancelation of both self and externally-generated low (<15 Hz) frequency spatially diffuse AM (i.e., first-order) stimuli (*Bastian et al., 2004*; *Bastian, 1999*; *Chacron, 2006*; *Chacron et al., 2005*; *Clarke and Maler, 2017*; *Bastian, 1986b*; *Bastian, 1996a*; *Bastian, 1996b*; *Bastian, 1998*), thereby allowing better detection of spatially localized stimuli (e.g., those caused by prey) (*Bastian et al., 2004*; *Litwin-Kumar et al., 2012*). Specifically, previous studies have shown that this pathway generates a negative image of the stimulus, thereby strongly reducing ELL pyramidal cell responses. More recent studies have shed light on how short-term burst-timing dependent depression helps in sculpting this negative image in an adaptive fashion (*Bol et al., 2011*; *Bol et al., 2013*; *Mejias et al., 2013*). In contrast, our results have demonstrated an important new function for indirect feedback in mediating optimized coding of envelope (i.e., second-order) stimuli by selectively attenuating responses to low frequencies.

How then can the same feedback pathway mediate attenuation of responses to both low-frequency AMs and envelopes? One possibility is that in both cases it is the same mechanisms that facilitate response attenuation. This is however unlikely to be the case because of the important difference between the frequency ranges of AMs and envelopes for which responses are attenuated, as mentioned above. Specifically, while the response to a 1 Hz AM will be strongly attenuated by feedback (*Bol et al., 2011*), our results show that this is not the case for a 1 Hz envelope. Previous results have shown that SK1 channels strongly determine envelope but not AM tuning properties. Specifically, pharmacological inactivation of SK1 channels gave rise to minimal effects on responses to AMs (*Toporikova and Chacron, 2009*) but compromised optimized coding of envelopes by causing the tuning curve to become broadband (*Huang et al., 2016*). In fact, the effects of indirect feedback inactivation and of SK1 channel antagonists on ELL pyramidal cell tuning to envelopes were strikingly similar (compare our *Figure 4B* to *Figure 6B* of (*Huang et al., 2016*)). We propose that indirect feedback excitation provides the necessary intracellular calcium entry that activates SK1 channels. These in turn give rise to spike frequency adaptation, thereby selectively attenuating responses to low envelope frequencies and causing the envelope tuning curve to become high-pass (*Huang et al., 2016*; *Benda and Herz, 2003*; *Deemyad et al., 2012*) (see (*Huang and Chacron, 2017*) for review). Thus, blocking indirect feedback input would strongly attenuate calcium entry via

NMDA receptors, thereby effectively inhibiting SK1 channels and explaining why the effects on ELL pyramidal cell tuning to envelopes are similar to those previously observed after application of SK channel antagonists (*Huang et al., 2016*). The fact that calcium-permeable NMDA receptors are highly expressed within the apical dendrites of ELL pyramidal cells (*Harvey-Girard and Dunn, 2003*; *Bottai et al., 1997*), where indirect feedback input terminates (*Berman and Maler, 1999*) and SK1 channels are located (*Ellis et al., 2008*), supports our hypothesis. However, the exact mechanisms by which SK1 channels determine responses to envelopes but not AMs, though critical in order to enable the indirect pathway to perform multiple functions, are not well-understood and should be the focus of future studies. We also note that anatomical studies have shown that multiple cell types within nP project to EGp (*Sas and Maler, 1983*; *Sas and Maler, 1987*). However, with the exception of multipolar cells, the electrophysiological properties of these neurons are unknown to this day. It is very likely that projections from these multiple types underlie the different functions of the indirect feedback pathway but further studies are needed to test this hypothesis.

It is important to note that the envelope stimuli considered here are behaviorally relevant and contain important information as to the relative position between both conspecifics. Since our results show that the indirect feedback did not attenuate responses to envelopes with higher (>0.2 Hz) frequencies, it is conceivable that these could interfere with the detection of other behaviorally-relevant low frequency AM stimuli (e.g., those caused by prey) since the responses to both stimuli should be enhanced by the direct feedback pathway. This is unlikely to be the case however because: 1) the AM stimuli caused by prey are spatially localized whereas envelope stimuli are spatially diffuse and; 2) the temporal frequency content of envelopes tends to be much lower (0 – 1 Hz) than that of prey stimuli (0 – 10 Hz) (*Stamper et al., 2013*; *Nelson and Maciver, 1999*). We hypothesize that these differences in spatial extent and temporal frequency content are used by the animal to distinguish between both stimuli, but further studies are needed to verify that this is the case.

We have shown for the first time how neurons within nP that project either directly or indirectly back to ELL respond to envelopes. Stellate cells projecting directly to ELL pyramidal cells displayed broadband tuning curves. The fact that ELL pyramidal cells displayed broadband tuning curves after pharmacological inactivation of indirect descending input suggests that they perform little filtering of input from stellate cells. Interestingly, multipolar cells projecting indirectly have tuning properties that are similar to those of ELL pyramidal cells and optimize coding via temporal whitening. This result has important implications for understanding how temporal whitening of envelopes occurs in ELL pyramidal cells as it implies that the output of the nP received by EGp granule cells that project back to ELL is temporally whitened. It is very likely that the output of nP multipolar cells, which is critical in determining ELL pyramidal cell tuning and responses to envelopes, undergoes significant filtering both by EGp granule cells to enable filtering by SK1 channels in ELL pyramidal cells. Further studies are needed to gain further understanding as to the underlying mechanisms.

Our results provide novel functional roles for the indirect feedback pathway in optimizing ELL pyramidal cell responses to envelopes through temporal whitening by selective attenuation of low frequencies. It should be noted that the architecture of the indirect feedback pathway by which parallel fibers emanating from the EGp terminate onto distal ELL pyramidal cell apical dendrites shares many similarities with the cerebellum and other cerebellum-like structures (e.g., the dorsal cochlear nucleus). Studies performed in cerebellum as well as cerebellum-like structures have shown clear common functions and mechanisms mediating how such descending input in attenuating or even cancelling responses to self-generated sensory input (*Cullen, 2011*; *Bell et al., 2008*; *Requarth and Sawtell, 2011*; *Warren and Sawtell, 2016*; *Sawtell, 2017*; *Singla et al., 2017*). It is likely that our results showing that feedback input from parallel fibers in the cerebellar-like ELL mediate optimized coding of natural stimuli through whitening will be applicable to the cerebellum as well as other cerebellum-like structures.

Finally, recent results have emphasized the critical role of descending pathways, which are found ubiquitously in sensory systems (*Cajal, 1909*; *Holländer, 1970*; *Ostapoff et al., 1990*; *Sherman and Guillery, 2002*), in determining accurate sensory neural and behavioral responses to sensory input (*Manita et al., 2015*; *Kwon et al., 2016*; *Takahashi et al., 2016*). In particular, it was shown that feedback terminating on the apical dendrites of cortical pyramidal neurons was essential in determining both responses and perception to somatosensory input (*Takahashi et al., 2016*). Critically, the electrosensory system of weakly electric fish displays many documented similarities with the mammalian visual, auditory, and vestibular systems (*Clarke et al., 2015*; *Chacron et al., 2011*;

*Metzen et al., 2015*). Thus, given that temporal whitening has been observed across sensory systems (*Dan et al., 1996*; *Wang et al., 2003*; *Pozzorini et al., 2013*; *Lundstrom et al., 2010*), it is likely that our results showing how feedback pathways mediate temporal whitening of sensory input by ELL pyramidal cells in the electrosensory system of weakly electric fish will be generally applicable to other systems.

## Materials and methods

### Animals

The wave-type weakly electric fish *Apteronotus leptorhynchus* was used exclusively in the present study. Fish were supplied by tropical fish suppliers and were housed in groups of 2 – 10 at appropriate water temperatures and water conductivities similar to their natural habitats according to published guidelines (*Hitschfeld et al., 2009*). All procedures were approved by McGill University's animal care committee under protocol 5285 and were performed in accordance to the guidelines set out by the Canadian Council of Animal Care.

### Surgery

0.1 – 0.5 mg of tubocurarine (Sigma) was injected intramuscularly in order to immobilize the fish for both electrophysiology and behavioral experiments. Experiments were performed in a tank (30 cm x 30 cm x 10 cm) filled with the fish's home tank water. The fish were respired using a constant flow of 10 mL/min of oxygenated water over its gills for the duration of the experiment. A 2 mm$^2$ hole was then exposed over either the hindbrain and/or midbrain near T0 to gain access to either ELL pyramidal neurons or nP neurons respectively for electrophysiology and/or drug injection. Bilateral exposure of the brain was performed for experiments requiring bilateral drug injections.

### Stimulation

The electric organ discharge of *A. leptorhynchus* is neurogenic, and therefore is not affected by injection of curare. All stimuli consisting of AMs of the animal's own EOD were produced by triggering a function generator to emit one cycle of a sine wave for each zero crossing of the EOD as done previously (*Bastian et al., 2002*). The frequency of the emitted sine wave was set slightly higher (30 – 40 Hz) than that of the EOD, which allowed the output of the function generator to be synchronized to the animal's discharge. The emitted sine wave was subsequently multiplied with the desired AM waveform (MT3 multiplier; Tucker Davis Technologies), and the resulting signal was isolated from ground (A395 linear stimulus isolator; World Precision Instruments). The isolated signal was then delivered through a pair of chloridized silver wire electrodes placed 15 cm away from the animal on either side of the recording tank perpendicular to the fish's rostro-caudal axis. Depending on polarity, the isolated signal either added or subtracted from the animal's own discharge. The resultant signals which arrives at the fish's skin was approximated using a dipole with 1 mm of distance between the two poles to simulate what the electroreceptors would pick up.

Both neural and behavioral experiments utilized stimuli consisting of a 5 – 15 Hz noise (4th order Butterworth) carrier waveform (i.e., AM) whose amplitude was further modulated (i.e., envelope) at frequencies ranging from 0.05 to 1 Hz, a behaviorally relevant range of frequencies which mimicked the envelope signals due to relative movement between two fish (*Stamper et al., 2013*; *Metzen and Chacron, 2014*). The depth of modulation for the envelope was approximately 20% of the baseline EOD amplitude as in previous studies (*Deemyad et al., 2013*; *Metzen et al., 2016*; *Metzen and Chacron, 2017*). This was confirmed using the dipole recording mentioned above.

We note that movement envelopes, which are the focus of the current study, are fundamentally different than so-called 'social' envelopes that are instead due to interaction between the EODs of three of more fish (*Stamper et al., 2013*) and which have been the focus of previous studies (*Middleton et al., 2006*; *Stamper et al., 2012*; *Savard et al., 2011*; *McGillivray et al., 2012*; *Thomas et al., 2018*). Indeed, for social envelopes, the frequency content of the envelope is mostly determined by the frequencies of the three EODs. This is because the envelope frequency is given by the difference between the two resulting beat frequencies. In contrast, for movement envelopes occurring during interactions between two conspecifics, the envelope frequency content is not determined by the two EOD frequencies or the resulting beat frequency. Rather, it is determined

solely by the relative movements of both fish (*Fotowat et al., 2013*; *Metzen and Chacron, 2014*). We further note that field studies have shown that Apteronotid species tend to encounter movement envelopes much more frequently than social envelopes: this is because they tend to be found in groups of 2 much more frequently than in groups of 3+ (*Stamper et al., 2010*). It is expected that social envelopes will be more relevant for weakly electric fish species that tend to be more gregarious (e.g., *Eigenmannia virescens*).

## Pharmacology

The composition of the vehicle/control saline was as follows: (all chemicals were obtained from Sigma): 111 mM NaCl, 2 mM KCl, 2 mM CaCl$_2$, 1 mM MgSO$_4$, 1 mM NaHCO$_3$ and 0.5 mM NaH$_2$PO$_4$. The pH of the saline solution was 6.8. Glutamate (Sigma), lidocaine (Astra Pharmaceuticals) and CNQX 6-cyano-7-nitroquinoxaline-2,3-dione (CNQX, Sigma) were dissolved in saline for application as done before (*Huang et al., 2016*; *Deemyad et al., 2013*). Drug application electrodes were made using either two-barrel KG-33 glass micropipettes (OD 1.5 mm, ID 0.86 mm, A-M Systems) pulled by a vertical micropipette puller (Stoelting Co.) or single barrel pipettes to a fine tip and subsequently broken to attain a tip diameter of ~5 µm for each barrel.

The two barrel pipettes were used for separate application of either lidocaine (1 mM) or CNQX (1 mM), as well as glutamate (1 mM) or saline. In order to block the indirect feedback, we injected CNQX into the ELL in proximity of a pyramidal neuron we were recording from, which we confirmed by using the excitatory response to glutamate application, as done previously (*Deemyad et al., 2013*). We also blocked the indirect feedback pathway by injecting lidocaine into the praeeminential electrosensory tract (PET) (behavior: bilateral; neurons: ipsilateral) projecting to the ipsilateral EGp rostral to ELL, as done previously (*Bastian, 1986b*). We then compared ELL neural and behavioral responses before and after injection. In order to block both the direct and indirect feedback pathways, we instead inserted two pipettes containing lidocaine into both the ipsilateral and contralateral nPs. Again, both ELL neural and behavioral responses were compared before and after injection. Bilateral injections of lidocaine were performed in order to completely silence the respective feedback pathways to directly observe the effect on neural responses and behavior, as done previously (*Huang et al., 2016*; *Deemyad et al., 2013*). In some cases, lidocaine was injected into the contralateral nP while recording from a pyramidal cell within the ipsilateral ELL (*Figure 4—figure supplement 1*). Saline controls were performed in the nP and we observed that there was no effect of the microinjection itself on either electrophysiology or behavior (*Figure 4—figure supplement 2*). All pharmacological injections were performed using a duration of 130 ms at 103 – 138 kPa using a Picospritzer (General Valve).

## Electrophysiology

We used well-established techniques to perform extracellular recordings with Woods metal electrodes from pyramidal cells (*Frank and Becker, 1964*) located within the lateral segment of the ELL based on recording depth and mediolateral placement of the electrode on the brain surface as done previously (*Huang and Chacron, 2016*; *Krahe et al., 2008*). We recorded pyramidal cells for control in conjunction with their responses after either lidocaine (nP injections: n = 7; PET injections: n = 9), CNQX (n = 8), or saline (n = 8) injections. In addition, by tailoring the tip shape of our Woods metal electrodes, we also performed extracellular recordings from nP stellate (n = 11) and multipolar cells (n = 8) in the midbrain. We confirmed the identity of each cell type based on recording depth, spontaneous baseline firing rates, and AM tuning curves (Figure 6 - figure supplement 1). While stellate cells have low spontaneous firing rates (1.37 ± 0.42 Hz) and do not respond to high AM frequencies, multipolar cells have high spontaneous firing rates (55.53 ± 8.05 Hz and do respond well to high AM frequencies > 32 Hz. These results match the results found in the literature (*Bastian and Bratton, 1990*; *Bratton and Bastian, 1990*), confirming our recordings were appropriate. All recordings were digitized at 10 kHz sampling rate using CED 1401 plus hardware and Spike2 software (Cambridge Electronic Design) and stored on a computer hard disk for offline analysis.

## Behavior

Animals were immobilized by an intramuscular injection of 0.1 – 0.5 mg tubocurarine and set up in the recording tank similarly to the method described above. Different surgeries were performed

depending on the pharmacology protocol. Both nPs or both ELLs were exposed on either side of the head in order to bilaterally inject lidocaine (nP injection: n = 9; PET injection: n = 8), saline (n = 8) or CNQX (n = 7), respectively. Pipettes containing lidocaine/saline were placed approximately 1000 – 1250 μm below the surface where the nP is located, while pipettes containing CNQX were placed superficially at about 200 – 300 μm below the surface of the hindbrain where the EGp feedback terminates on the apical dendrites of pyramidal cells. Additionally, lidocaine injections were performed 50 – 100 μm below the brain surface adjacent to the rostral end of the ELL, where the PET is located terminating on the EGp, in order to block indirect feedback. Multiple injections (typically 3 – 5) were performed to ensure that both hemispheres of nP and ELL were sufficiently affected by the pharmacological agents. Stimuli were then presented as described above in order to elicit behavioral responses before and after drug application. The animal's behavior was recorded through a pair of electrodes located at the rostrum and tail of the animal. The zero-crossings of the recorded EOD signal were used to generate a binary sequence as described above that was low-pass filtered (2nd order Butterworth filter with 0.05 Hz cut-off frequency) to obtain the time-varying EOD frequency.

## Data analysis

Data obtained from ELL pyramidal neurons were pooled as there is no difference in envelope response between ON- and OFF-type pyramidal cells (*Huang and Chacron, 2016*). All data analysis was performed offline using custom written codes in MATLAB software (MathWorks) (*Huang et al., 2018*). The recorded potentials were first high-pass filtered (100 Hz; 8th order Butterworth). Spike times were sorted using Spike2 software and defined as the times at which the signal crossed a given threshold value from below. To quantify the neural responses and relate them to the stimulus envelope, we used linear systems identifications techniques to compute the gain relationships to envelope frequency. We approximated the gain by averaging over the cycles of the stimulus and fitting a sinewave to the resultant cycle histogram to determine the firing rate modulation. We then divided the amplitude of the firing rate modulation to the stimulus envelope amplitude observed in the dipole to obtain the gain to any given envelope frequency. Our filtered firing rates were obtained using a second-order Butterworth filter with cutoff frequencies of 0.2, 0.35, 0.75, 1.5, 2.5, and 3.5 Hz for envelope frequencies 0.05, 0.1, 0.2, 0.5, 0.75, and 1 Hz, respectively, as done in previous studies (*Huang and Chacron, 2016*). Gain values calculated for behavior was performed using similar methods.

Responses to AMs were calculated using standard techniques by determining the spike-triggered average (STA) change in amplitude of the AM stimuli. The STA is the mean stimulus waveform that triggers an action potential and was obtained by averaging the stimulus waveforms within a 2 s time window surrounding each spike. We used the same envelope stimuli containing 5 – 15 Hz AM in order to calculate our STAs and determine the response to AMs. The response was then quantified as the peak-to-peak amplitude of each STA change in amplitude and was compared before and after drug application. We then quantified the change before and after drug application as a percentage of control, where the control STA was normalized to 100%.

Temporal whitening performed by ELL pyramidal neurons and nP neurons were calculated based on their observed tuning properties by squaring the gain at a given envelope frequency and multiplying it by the power of the natural stimulus whose spectrum decays as a power law with exponent $\alpha = -0.8$. The result provided us with an accurate estimation of the response power of the neuron across the range of frequencies we investigated. Indeed, previous studies have shown that an accurate approximation of the response power spectrum could be correctly predicted using the tuning function with this transfer function, as a change in the tuning curve of the neuron directly caused changes in the response power experimentally (*Huang et al., 2016*; *Huang and Chacron, 2016*). From the temporal whitening response power curves, the whitening index was calculated by taking the area under the power spectrum curve of the spiking response using a trapezoidal method and dividing by that obtained by replacing all values by the maximum value in the power spectrum. The whitening index ranges between 0 and 1, where one indicates complete whitening (i.e., a power spectrum that is independent of temporal frequency), as done previously (*Huang et al., 2016*). Finally, the change in sensitivity between control and/or drug conditions was calculated using the following formula:

$$\frac{G_{Drug} - G_{Control}}{G_{Control}} \times 100$$

where $G_{Drug}$ is the gain response of the neuron after drug injection at a given envelope frequency and $G_{Control}$ is the gain response of the neuron under control conditions at the same given envelope frequency. We then pooled the changes in sensitivity across envelope frequencies in order to obtain the case individually for neuron and behavior.

## Statistics

Statistical significance was assessed through a non-parametric Kruskal-Wallis ANOVA test if the data was unpaired or Wilcoxon signed-rank test for paired measures at the $p=0.05$ level. Data is presented as mean ± standard error (SEM). For the whisker boxplot in Figure 6 - figure supplement 1, the central mark indicates the median, and the bottom and top edges of the box indicate the 25th and 75th percentiles, respectively. The whiskers extend to the range of data points.

## Additional information

### Funding

| Funder | Author |
|---|---|
| Canadian Institutes of Health Research | Maurice J Chacron |
| Fonds de Recherche du Québec - Nature et Technologies | Maurice J Chacron |

The funders had no role in study design, data collection and interpretation, or the decision to submit the work for publication.

### Author contributions

Chengjie G Huang, Conceptualization, Data curation, Formal analysis, Validation, Investigation, Visualization, Methodology, Writing—original draft, Writing—review and editing; Michael G Metzen, Conceptualization, Formal analysis, Validation, Investigation, Visualization, Methodology, Writing—original draft, Writing—review and editing; Maurice J Chacron, Conceptualization, Resources, Software, Supervision, Funding acquisition, Validation, Writing—original draft, Project administration, Writing—review and editing

### Author ORCIDs

Chengjie G Huang (iD) https://orcid.org/0000-0001-7491-6060
Michael G Metzen (iD) http://orcid.org/0000-0002-2365-4192
Maurice J Chacron (iD) http://orcid.org/0000-0002-3032-452X

### Ethics

Animal experimentation: All procedures were approved by McGill University's animal care committee and were performed in accordance to the guidelines set out by the Canadian Council of Animal Care. Protocol 5285.

### Decision letter and Author response

Decision letter https://doi.org/10.7554/eLife.38935.019
Author response https://doi.org/10.7554/eLife.38935.020

## Additional files

### Supplementary files

• Transparent reporting form

DOI: https://doi.org/10.7554/eLife.38935.015

**Data availability**

All data have been deposited on Figshare under the URL: https://figshare.com/s/b9e094a67a38e29212e8

The following dataset was generated:

| Author(s) | Year | Dataset title | Dataset URL | Database, license, and accessibility information |
|-----------|------|---------------|-------------|--------------------------------------------------|
| Huang CG, Metzen MG, Chacron MJ | 2018 | Figure source data for: Feedback optimizes neural coding and perception of natural stimuli | https://figshare.com/s/b9e094a67a38e29212e8 | Available at figshare under a CC0 Public Domain licence (https://figshare.com/). |

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
