## [Decision Letter]

Thank you for submitting your article "Feedback optimizes neural coding and perception of natural stimuli" for consideration by *eLife*. Your article has been reviewed by three peer reviewers, including Catherine Emily Carr as the Reviewing Editor and Reviewer #1, and the evaluation has been overseen by Eve Marder as the Senior Editor. The following individual involved in review of your submission has agreed to reveal their identity: Len Maler (Reviewer #2).

The reviewers have discussed the reviews with one another and the Reviewing Editor has drafted this decision to help you prepare a revised submission.

Summary:

The main strength of the manuscript is the insight into mechanisms whereby sensory neurons match their tuning properties to natural stimulus statistics.

The manuscript extends previous published work on the role of feedback to the first order electrosensory lobe (ELL) of a gymnotiform weakly electric fish. Previous work emphasized the response of ELL pyramidal cells to first order modulations (AMs) of the amplitude of the electric field produced by the sinusoidal electric organ discharge (EOD). A direct feedback pathway enhances responses to local AMs, i.e., those produced by objects causing AMs over a restricted portion of the body surface. The indirect feedback pathway attenuates the responses to the global low frequency AMs and thus prevents interference with prey detection. The novel contribution of this manuscript is that both feedback pathways also control the pyramidal cell responses to envelope signals. This is surprising because, not only are such signals detected via different neural mechanisms than AM signals, but their frequency content is mostly very low. The authors also show the effects on behavior and record from the cells that give rise to the direct and indirect pathways.

Essential revisions:

1) The major issues with the manuscript appear to be with presentation. As one reviewer wrote "if the Introduction clearly (and simply) gave the relevant background, the rest of the manuscript would be much easier to follow."

2) The regular and supplemental figures are intermixed (and sometimes repetitive). There is a lot of overlap between panels of different figures. Often the differences are difficult to see at first, it appears that at least three figures have half of the panels in common. Only after careful consideration one finds subtle differences between panels. The figures should be carefully re-evaluated.

3) It would be good to provide a concise summary of the anatomy rather than adding circuit descriptions piece by piece. For example, the authors mention that there are two different groups of nP neurons in the last paragraph of the Introduction, the notation for nP is clarified, in the first paragraph of the subsection “Descending input shapes neural responses and perception of envelopes”, as referring to the nuclei praeeminentialis, and the specific cell types are introduced in the subsection “Responses of nP neurons that give rise to descending input onto ELL pyramidal cells”. Also, more specifically in the sentences "the former displayed broadband envelope frequency tuning curves.…" and "the latter displayed high-pass envelope frequency tuning curves.…" it is ambiguous what former/latter refers to, because it could refer to nP/ELL cells or cells providing direct/indirect feedback.

4) The Introduction meanders between the discussion of how envelopes are represented, the contribution of SK1 channel, and feedback. The properties of SK1 channels are discussed in the Introduction but the interaction with feedback is only explained in the Discussion. The Introduction just sets them up as different and unrelated mechanisms for implementing optimal coding. This initially lends the impression that the present study contradicts previous published works by the authors.

5) A reviewer points out that stellate cells may be the only source of the direct feedback pathway. On the other hand, there are a number of nP neurons that project to the EGp and therefore contribute electrosensory input to the indirect feedback pathway. The authors should make clear that there are multiple nP cells that project to EGp and that therefore will contribute to the many effects of this complex feedback system. The multipolar cell does not respond very effectively to either low frequency global signals or object motion (local stimuli) and, as the authors show, it is well suited for modulation of envelope responses. The authors could put in one or two simple sentences just to make clear that the various functions of the indirect feedback pathway may be based on the multiplicity of nP cells projecting to EGp.

---

## [Author Response]

Essential revisions:1) The major issues with the manuscript appear to be with presentation. As one reviewer wrote "if the Introduction clearly (and simply) gave the relevant background, the rest of the manuscript would be much easier to follow."

We have made significant changes to the manuscript figures and text to improve presentation. Moreover, the Introduction has been reworked in order to give the relevant background clearly as requested.

2) The regular and supplemental figures are intermixed (and sometimes repetitive). There is a lot of overlap between panels of different figures. Often the differences are difficult to see at first, it appears that at least three figures have half of the panels in common. Only after careful consideration one finds subtle differences between panels. The figures should be carefully re-evaluated.

We understand the point and have made extensive revisions to the figures in order to clarify the presentation. Overall, the different manipulations are now all presented in Figure 2 and each subsequent figure clearly indicates which manipulation was performed. We have also harmonized the color scheme such that all data obtained after complete feedback inactivation are now shown in purple whereas all data obtained after indirect feedback inactivation are now shown in orange throughout. Moreover, we now refer either to “envelope frequency” or “AM frequency” throughout the manuscript to avoid confusion.

3) It would be good to provide a concise summary of the anatomy rather than adding circuit descriptions piece by piece. For example, the authors mention that there are two different groups of nP neurons in the last paragraph of the Introduction, the notation for nP is clarified, in the first paragraph of the subsection “Descending input shapes neural responses and perception of envelopes”, as referring to the nuclei praeeminentialis, and the specific cell types are introduced in the subsection “Responses of nP neurons that give rise to descending input onto ELL pyramidal cells”. Also, more specifically in the sentences "the former displayed broadband envelope frequency tuning curves.…" and "the latter displayed high-pass envelope frequency tuning curves.…" it is ambiguous what former/latter refers to, because it could refer to nP/ELL cells or cells providing direct/indirect feedback.

We have added a circuit diagram to Figure 1 and now describe the circuitry in detail at the beginning of the Results. We have also replaced “former” and “latter” by “nP neurons projecting directly to ELL” and “nP neurons projecting indirectly to ELL” in the sentence to avoid confusion.

4) The Introduction meanders between the discussion of how envelopes are represented, the contribution of SK1 channel, and feedback. The properties of SK1 channels are discussed in the Introduction but the interaction with feedback is only explained in the Discussion. The Introduction just sets them up as different and unrelated mechanisms for implementing optimal coding. This initially lends the impression that the present study contradicts previous published works by the authors.

We agree and now mention SK1 channels in the Discussion in order to better relate the results of the current manuscript to our previous ones.

5) A reviewer points out that stellate cells may be the only source of the direct feedback pathway. On the other hand, there are a number of nP neurons that project to the EGp and therefore contribute electrosensory input to the indirect feedback pathway. The authors should make clear that there are multiple nP cells that project to EGp and that therefore will contribute to the many effects of this complex feedback system. The multipolar cell does not respond very effectively to either low frequency global signals or object motion (local stimuli) and, as the authors show, it is well suited for modulation of envelope responses. The authors could put in one or two simple sentences just to make clear that the various functions of the indirect feedback pathway may be based on the multiplicity of nP cells projecting to EGp.

We agree with the reviewer and now mention in the Discussion as well as in the caption of Figure 1 that several other cell types within nP project to the EGP. These most likely underlie the different functional roles of this feedback pathway.